# Emergency Management of Electrical Storm: A Practical Overview

**DOI:** 10.3390/medicina59020405

**Published:** 2023-02-19

**Authors:** Fabrizio Guarracini, Eleonora Bonvicini, Sofia Zanon, Marta Martin, Giulia Casagranda, Marianna Mochen, Alessio Coser, Silvia Quintarelli, Stefano Branzoli, Patrizio Mazzone, Roberto Bonmassari, Massimiliano Marini

**Affiliations:** 1Department of Cardiology, S. Chiara Hospital, 38122 Trento, Italy; 2Department of Cardiology, University of Verona, 37126 Verona, Italy; 3Department of Radiology, Santa Chiara Hospital, 38122 Trento, Italy; 4Cardiac Surgery Unit, Santa Chiara Hospital, 38122 Trento, Italy; 5Department of Cardiac Surgery, Universitair Ziekenhuis Brussel-Vrije Universiteit Brussel, 1090 Brussels, Belgium; 6Cardiothoracovascular Department, Electrophysiology Unit, Niguarda Hospital, 20162 Milan, Italy; 7Heart Rhythm Management Centre, Universitair Ziekenhuis Brussel-Vrije Universiteit Brussel, European Reference Networks Guard-Heart, 1090 Brussel, Belgium

**Keywords:** electrical storm, ventricular arrhythmias, implantable cardioverter defibrillator, antiarrhythmic drugs, medical emergency

## Abstract

Electrical storm is a medical emergency characterized by ventricular arrythmia recurrence that can lead to hemodynamic instability. The incidence of this clinical condition is rising, mainly in implantable cardioverter defibrillator patients, and its prognosis is often poor. Early acknowledgment, management and treatment have a key role in reducing mortality in the acute phase and improving the quality of life of these patients. In an emergency setting, several measures can be employed. Anti-arrhythmic drugs, based on the underlying disease, are often the first step to control the arrhythmic burden; besides that, new therapeutic strategies have been developed with high efficacy, such as deep sedation, early catheter ablation, neuraxial modulation and mechanical hemodynamic support. The aim of this review is to provide practical indications for the management of electrical storm in acute settings.

## 1. Introduction

### 1.1. Definition

Electrical storm (ES) is a medical emergency characterized by electrical instability and recurrence of clustered ventricular arrythmias (VAs) within a short amount of time, which frequently can lead to hemodynamic instability. The current definition according to the latest European Society of Cardiology (ESC) guidelines is the occurrence of sustained VAs three or more times within 24 h requiring intervention, with each event separated by at least five minutes [1]. In case of the presence of an implantable cardioverter defibrillator (ICD), ES is defined as three or more separate ventricular tachycardia (VT) or ventricular fibrillation (VF) episodes leading to (ICD) therapies within 24 h, either anti-tachycardia pacing (ATP) or cardioversion/defibrillation, thus excluding arrythmias under the device detection threshold or spontaneously terminated [2,3].

### 1.2. Incidence, Basic Epidemiological Aspects and Risk Factors

ES in patients implanted with an ICD has an incidence of about 10–20% in secondary prevention implantations, while a lower incidence is present in primary prevention implantations [3]. The incidence of ES appears to be higher in structural heart disease or inherited channelopathies, with comparable rates between ischemic cardiomyopathy (ICM) and nonischemic cardiomyopathy (NICM) frequently related to the common presence of fibrotic arrhythmic substrate [4]. Structural heart diseases involve remodeling, widespread fibrosis, fatty deposition and discrete scars that, as a consequence, change electrophysiological properties through ion channel remodeling [5]. ES onset is typically preceded more often by monomorphic VT than polymorphic VT or VF (Odds Ratio (OR) 1.79; 95%, Confidence Interval (CI) 1.35–2.39) [6,7]. Monomorphic VT originates from reentry due to an electrophysiological substrate, which could trigger and frequently sustain the ES [8]. In a meta-analysis of VA storm studies, 94% of ES occurred in structural heart diseases (mostly ischemic and non-ischemic) and were initiated by monomorphic VT in 80% of storms with a scar-related reentrant mechanism [9]. When polymorphic VT or VF is the initiator of ES generally, this clinical emergency is frequently related to acute myocardial infarction (MI) and advanced heart failure (HF). This clinical scenario is also often complicated by a critical hemodynamic instability in different underlying cardiac diseases [10].

Previous history of VAs, as in the context of ICD implantation for secondary prevention, represents the strongest predictor of ES (OR 3.62; 95% CI 1.08–12.14) [6,11].

This can be explained by the detection capability of ICDs and their therapeutic role, necessary to treat VAs appropriately, and by the predisposition of the patients to suffering a new arrhythmic recurrence after the first episode [7,11,12,13]. Patients expected to be at higher risk of ES are those with a severe reduction of ejection fraction (EF 23.08%; 95% CI 25.56 to 0.62) [6]. There is no absolute agreement about the role of left ventricular ejection fraction (EF) as a risk factor for ES; however, according to data in literature, ES patients have, in general, an EF reduction of 3% compared to patients without ES, in a population with already poor ejection fraction [3,12,13,14]. Some drugs have been associated with a five-fold increased risk of ES, such as class I antiarrhythmic drugs (AADs) (OR 5.20; 95% CI 1.35–20.01), whereas amiodarone and beta-blockers have not been significantly associated with a higher risk [6]. This risk is probably related to class I (AADs) proarrhythmic effects, especially in the presence of structural heart disease [15]. Poor medication adherence is also a risk factor for ES [4].

Statistically relevant risk factors such as secondary prevention ICD implantation, lower EF, monomorphic VT and class I AAD may be used to identify populations at higher risk of developing ES [6].

### 1.3. Prognosis

ES is considered a significant predictor of mortality, irrespective of the underlying heart disease and the history of previous VT/VF [6]. According to metanalysis data, ES accounted for a nearly three-fold increased risk of death (Relative Risk (RR) 3.15; 95% CI 2.22–4.48), which remains significant, comparing ES patients to patients with previous history of one or more unclustered VTs or VFs (RR 2.51; 95% CI 1.38–4.58) [6]. ES is also associated with an increased combined risk of death, cardiac transplantation and hospital admission for acute decompensated HF or cardiogenic shock (RR 3.39; 95% CI 2.31–4.97) [6]. The prognosis of patients with ES is directly related to the underlying substrate; however, ES occurring in patients with structurally normal hearts is associated with a better outcome [1,16].

The burden of VAs is related to sudden cardiac death rates in ICD patients [17]: it might be a consequence of either acute severe left ventricular dysfunction and cardiogenic shock due to VA recurrence [18] or due to multiple shocks [19,20]. In fact, termination of VAs by shocks leads to higher mortality rates (20% increased risk per shock episode) compared to any therapy or anti-tachycardia pacing (ATP) [10,21], although even VT storm terminated with ATP is also associated with increased mortality [21].

Moreover, according to the AVID (this is the name of the trial) and MADIT-II trials, the risk of death is highest 3 months after the ES event, it is commonly associated with non-sudden cardiac causes, and 4% of patients develop ES on average at 20.6 months of follow-up [13,22].

Catheter ablation (CA) of ES has demonstrated a clear reduction of VA recurrence, supporting the hypothesis of cardiac mortality reduction by preventing systolic dysfunction through the reduction of VA burden [23,24].

## 2. Causes

ES triggers should be researched but are often not identifiable and are not fully understood. Probably, the development of ES derives from the superimposition of a pre-existing vulnerable substrate and disturbing factors such as autonomic tone, heart rate or myocardial ischemia (Table 1).

ES might be a manifestation of MI, ICM and NICM and inherited arrhythmia syndromes such as Brugada syndrome (BrS), early repolarization syndrome (ERS), short-coupled ventricular fibrillation (SCVF), long QT syndrome (LQTS), short QT syndrome (SQTS), and catecholaminergic polymorphic ventricular tachycardia (CPVT). Usually, ES occurs in conditions of elevated sympathetic tone, but sometimes even parasympathetic tone might be involved. Focal excitation due to abnormal automaticity or triggered activity might become a trigger that initiates or maintain reentrant VT. Purkinje fibers are considered possible trigger sources for polymorphic VT or VF storm in patients with MI [25] and HF [26], and they might be susceptible to CA [27].

Electrical remodeling and abnormal impulse propagation are possible sources of ES; in fact, as suggested by reports in the literature, the combination of an EF < 25% and a wide ECG QRS (duration > 120 msec) is a strong predictor of ES [28]. Cardiac dysfunction is related to ES development; in fact, cardiac resynchronization therapy (CRT) reduces VAs [29], and CRT non-responders experience more ventricular arrhythmia events than responders [30].

In structurally normal hearts with occurrence of idiopathic VT, the underlying mechanism frequently might be delayed after depolarization-mediated ectopy in outflow tract or papillary muscles [31].

## 3. Early Evaluation and Acute Management

The management of ES requires a multidisciplinary approach to ensure an effective treatment. Acute management aims at stopping ES and suppressing arrhythmogenic stimuli in order to prevent severe consequences such as cardiogenic shock. More definitive management targets the underlying VA substrate to terminate and prevent arrhythmia recurrence. Poorly tolerated VT with a short cycle length or in patients with poor cardiac reserve can present with syncope, cardiac arrest or cardiogenic shock. In that setting, emergent stabilization, advanced cardiac life support and immediate cardioversion are mandatory. Prompt termination of VT is recommended even for those tolerated, because rapid hemodynamic deterioration may occur.

If hemodynamic support is not necessary, patients often experience palpitations, dizziness, HF, chest pain or ICD shocks. In that situation, the identification and treatment of potential triggers and attempts to interrupt VAs with pharmacotherapy are recommended as a first approach [10,35] (Figure 1).

### 3.1. ECG 12 Lead Evaluation

Documentation of any hemodynamically tolerated wide QRS tachycardia on 12-lead ECG is important: firstly, to exclude supraventricular tachycardia, and then to confirm the diagnosis of ventricular tachycardia and to differentiate the type of VT.

Pre-excited atrial fibrillation needs to be excluded with the recognition of an "FBI" (fast, broad, irregular) ECG pattern [36], and in this case, intravenous administration of drugs that slow AV conduction should be avoided. In other cases, administration of adenosine, vagal measures, or beta-blockers with continuous recording of 12-lead ECG is useful to identify supraventricular tachycardia, but in some cases, intravenous adenosine may also interrupt specific VT subtypes that are triggered by the support of cyclic adenosine monophosphate (cAMP).

A 12-lead ECG has an important role in defining the type of arrythmia (monomorphic or polymorphic) and the arrythmia’s anatomic origin site.

### 3.2. Reversible Causes Treatment

Reversible causes of ES may account for up to 50% of acute coronary syndrome [37,38] or myocardial ischemia (either coronary spasm or thrombosis); myocardial ischemia should be promptly treated with invasive angiography and coronary angioplasty if necessary.

Triggers for ES should be sought but are often unidentifiable. In the SHIELD trial, a precipitating cause could be identified in only 13% of electrical storms, typically represented by worsening heart failure, which should be treated in favor of better hemodynamic stability, and electrolyte disturbances [3]. Electrolyte abnormalities should be corrected, especially potassium [39,40,41] and magnesium serum levels: their deficiency may be associated with Torsade de Pointes (TdP), and intravenous magnesium might be effective even if magnesium serum levels are within range [42]. QT prolongation is often associated with recurrence of TdP, and in this setting, bradycardia may become a trigger factor: in fact, increasing cardiac frequency through beta adrenergic receptor stimulation (with isoproterenol) or transvenous pacing has proved to be effective.

The causal role of AADs should be suspected in patients being treated with agents known to alter the electrical properties of the heart (with QRS or QT prolongation) or cause electrolyte abnormalities (as diuretics). Potentially responsible drugs must be discontinued whenever drug-related arrythmia is suspected [40,43], and caution is needed in subsequent anti-arrhythmic therapy administration [44].

Sympathetic nervous system activation is a significant modulatory factor [10] that contributes to the initiation and maintenance of ES [45]: sympathetic inhibition plays an important role, especially in subsets of patients with congenital long QT syndrome (LQTS), catecholaminergic polymorphic VT (CPVT) [46,47] and pheochromocytoma [48]. Some studies supported the role of sympathetic activation, reporting data of higher prevalence of ES on working days and during daytime hours [49], and even related to stressful events [50]. Sympathetic inhibition could be achieved by pharmacologic receptor wide suppression using non-selective beta blockers, by deep sedation or cardiac sympathetic denervation. In contrast, arrhythmia onset in particular settings such as BrS and ERS, SQTS and SCVF usually occur during sleep or rest, conditions associated with elevated parasympathetic tone, in which isoproterenol might be effective in suppressing arrythmia recurrence [51,52,53].

Other potentially triggering factors are fever, acute starvation and dieting [54].

### 3.3. Device (ICD) Interrogation and Reprogramming

In patients with ICDs, ES often presents with recurrent ICD ATPs or shocks that frequently lead to myocardial distress and can strengthen the adrenergic response, which can perpetuate ventricular arrhythmias. ICD reprogramming is essential to avoid inappropriate treatment of supraventricular arrhythmias or to detect lead damage, avoiding shocks [1]. In particular manual overdrive pacing may terminate VTs with a cycle length under the programmed ICD detection rate, and magnet use over the device pocket can disable therapy in complex acute clinical scenarios [1,35].

## 4. Medical Therapy

### 4.1. Sodium-Channel Blockers (Class I)

Sodium-channel blockers are a group of drugs that act by the inhibition of sodium influx through membranes that implies shortening of initial rapid depolarization, reduction of cell excitability and conduction velocity. Class I drugs are often used in ES, but patients need to be strictly monitored because of their negatively inotropic effects, which can determine more heart failure and arrhythmic episodes (Table 2).

Class Ia (Quinidine)

Quinidine is a class IA AAD that has the capacity of blocking the transient outward potassium current [I_T0_]. It has been recommended as a first line treatment for ES in Brugada syndrome. Good results have been seen also in SQTS, ERS or J-wave syndromes [31,55,56,57]. Dan L Lin et al. conducted a retrospective analysis of 37 patients who were referred to their tertiary referral center after other therapies failed for VA suppression. These patients initiated oral quinidine, which reduced acute VAs from a median of three episodes during median of 3 days before and 4 days after quinidine initiation [58]. Quinidine has been proved to be of value for polymorphic VTs not responsive to other drugs in patients with coronary heart disease with or without acute ischemia [59].

Quinidine supplies are not always in stock, because while quinidine is usually available for the treatment of malaria, it is not available in many countries. Like all other Class I drugs, contraindications are previous MI, structural heart disease, and long QT syndromes. Adverse effects are frequent and must be monitored: gastrointestinal, hematologic, auditory and visual disturbances can be observed.

Class Ib (Mexiletine and Lidocaine)

Mexiletine and lidocaine are class IB antiarrhythmic agents that have efficacy in interrupting VAs in acute coronary syndrome. In this situation, altered membrane potential and pH reduction increase the drug-binding rate, causing better efficacy.

Lidocaine is effective in interrupting VAs principally related to acute myocardial ischemia [60]; instead, mexiletine was reported to be useful in arrythmias when it is added to the baseline antiarrhythmic therapy [61]. Sobiech et al. performed a retrospective cohort analysis of 17 patients treated with mexiletine for recurrent VT and/or VF, using patients themselves as self-controls. It was demonstrated that treatment with mexiletine significantly reduced the number of ES events (14 episodes vs. two episodes) and ICD interventions (317 interventions vs. nine interventions), in comparison with a matched period before initiation of treatment [62]. Lidocaine must be used with a reduced dose when there is a reduced liver blood flow, such as in shock, beta-blockade and severe HF. It is demonstrated to be more effective with high potassium levels.

Mexiletine has, moreover, been recognized for its role in suppression of ventricular arrythmias in LQTS [63]. Known side effects of class IB AADs are dose-dependent, and the major ones known are sinoatrial arrest, central nervous system impairment and, regarding mexiletine, also gastrointestinal complaints.

Class Ic (Procainamide, Flecainide and Propafenone)

Procainamide is a class IC AAD that can block the fast sodium channels and, therefore, it inhibits recovery after repolarization, prolongs the action potential and reduces the speed of impulse conduction. [64,65].

In the PROCAMIO study, intravenous procainamide and amiodarone were compared for the treatment of acute sustained monomorphic well-tolerated ventricular tachycardias: procainamide therapy was associated with fewer major cardiac adverse events and a higher proportion of tachycardia termination within 40 min [66]. Further studies are needed for its use as chronic therapy.

Contraindications are severe intraventricular conduction disturbance, reduced EF and BrS.

The other class IC AADs (Flecainide and Propafenone) are mainly used for supraventricular arrythmias. An exception is their administration in catecholaminergic polymorphic ventricular tachycardia, because of their negative effect on the RYR2 calcium channel and sodium currents [67].

### 4.2. Beta Blockers (Class II)

Beta blockers are competitive antagonists that block the receptor sites for endogenous catecholamines on adrenergic beta receptors of the sympathetic nervous system. An enhanced sympathetic tone has a predominant role in the triggering and maintenance of an ES. In this situation, suppression of the sympathetic tone is an important cornerstone for the management of the ES [68] (Table 1).

Non-selective beta blockers are usually preferred. Often, these patients have chronic HF or an ischemic heart disease that implies a downregulation of beta1 receptors and up-regulation of beta2 receptors. Furthermore, beta2-receptor activation induces hypokalemia, often related to arrythmia generation [69], and increases QT interval and dispersion of repolarization in the ventricular myocardium. Therefore, a non-selective beta blocker, such as propranolol, is frequently used in ES [70].

In a prospectively designed study ended in 2016, Chatzidou et al. randomly assigned therapy with either propranolol or metoprolol, combined with IV amiodarone, to ICD patients with ES. It was demonstrated that intravenous (IV) amiodarone with oral propranolol was safer and superior to IV amiodarone with oral metoprolol [71].

Short acting beta blockers, such as esmolol [72], or ultra-short-acting beta1-selective blockers, such as landiolol [73], could be used in ES, particularly in a compromised hemodynamic situation.

The J-Land II Study was an open-label uncontrolled multicenter study in which 29 patients with recurrent hemodynamically unstable VT or VF refractory to Class III antiarrhythmic drugs were treated with continuous intravenous infusion of landiolol. The efficacy endpoints were the proportion of patients free from recurrent VT/VF, the number of recurrent VT/VF events and the survival rate after 30 days. The study confirmed the efficacy for patients who did not respond to class III AADs, but further studies must be conducted in order to understand if this drug can be a first treatment choice [74]. Furthermore, beta blockers are the chosen medical treatment for some channelopathies, such as catecholaminergic polymorphic ventricular tachycardia.

Side effects such as excessive bradycardia, AV block and negative inotrope effect should be taken into account when using beta blockers. Contraindications to beta blocker use are severe sinus node dysfunction, coronary vasospasm, Brugada syndrome and severe asthma.

### 4.3. Amiodarone and Sotalol (Class III)

Amiodarone performs as a class III drug by blocking potassium currents during the third phase of the cardiac cycle, increasing the duration of the action potential as well as the effective refractory period for myocytes. Mostly when administered intravenously, amiodarone acts like other AADs: in fact, it is established that it can inhibit sodium channels (Class I) and block L-type calcium channels (Class II) and sympathetic receptors (Class IV).

With these characteristics, amiodarone is one of the most effective and used drugs in ES treatment [75,76]. It can be used either in hemodynamically stable situations or in unstable ones as part of ALS before other kinds of interventions. In addition to that, it could be used in hearts with structural disease when class IC AADs are usually contraindicated.

Even in patients in chronic therapy with amiodarone, loading could be useful in reducing the time needed for arrythmia control.

In acute settings and for brief periods of time, amiodarone could be used with great benefits and only small side effects, but long-term administration must be weighed with potential drug toxicity. Frequent extracardiac side effects are dysthyroidism, pulmonary fibrosis, hepatotoxicity and photosensitivity. Additionally, cardiac side effects should be taken into account: in fact, amiodarone can increase defibrillation thresholds [77] and in rare cases can cause or contribute to acquired long QT arrhythmia; hence, it is recommended to pay attention to amiodarone’s administration in prolonged QT intervals [78].

In this scenario, frequent evaluations of its toxicity must be performed, and it is recommended that amiodarone be used as a bridge to a more definitive treatment, such as catheter ablation.

Sotalol is another class III AAD; it inhibits rapid potassium channels, but in addition to that, has inhibitory effects on beta-1 adrenoreceptors with the effect of prolonging repolarization and increasing the refractory period. Sotalol has been reported to prevent both appropriate and inappropriate ICD shocks, with more efficacy in patients with MI [79,80]. It is important to consider that sotalol has only been studied in isolated VT tachycardias, and not in electrical storm; therefore, its efficacy has still to be defined.

Contraindications are severe depressed left ventricular dysfunction, significant left ventricular hypertrophy, creatinine clearance under 30 mL/min, coronary vasospasm and long QT syndromes. Particular attention must be paid in patients with concomitant treatment associated with QT interval prolongation and in hypokalemia status. A 2% risk of developing Torsade de Pointes is present, and close monitoring of the QT interval and creatinine clearance is needed [81].

Class III AADs were compared in the OPTIC study, a randomized controlled trial in which 412 patients were randomized to treatment for 1 year with amiodarone plus beta blocker, sotalol alone, or beta blocker alone and had ICD shock as primary outcome. It was proved that sotalol determines a reduction in the risk of shock in comparison with beta blocker alone, whereas amiodarone plus beta blocker was the most effective, but had an increased risk of drug-related adverse effects [82].

### 4.4. Calcium Channel Blockers (Class IV)

Verapamil and diltiazem are class IV AADs. These drugs depress calcium-dependent action potentials in slow-channel tissues and thus decrease the rate of automaticity, slow conduction velocity, and prolong refractoriness. They are not drugs typically used in ES.

Only verapamil was proved to be useful in some particular situations: in fact, it has been proposed as a therapy for the short-coupled variant of Torsade de Pointes, a disease that cannot be distinguished from idiopathic VF [83].

Nevertheless, current guidelines do not recommend the use of verapamil in broad QRS complex tachycardia of unknown mechanism [1].

### 4.5. Alternative Drugs

Isoproterenol is a beta agonist that in most conditions is considered proarrhythmic. It acts by increasing the calcium currents and stabilizing the epicardial action potentials, reducing the transmural heterogenicity. Because of these proprieties, it decreases elevated ST and suppresses VTs, avoiding the genesis of premature beats; therefore, it is demonstrated to be the first-line therapy in suppressing VT in patients with BrS [31,57,84].

Isoproterenol has been demonstrated to be useful in ES in idiopathic VF [85], ERS [86], TdP and beta blocker overdose. Studies has been conducted with positive results in acquired LQTS.

Isoproterenol must not be used in acute coronary syndromes. Adverse effects (such as headache, sweating and tremor) are very limited during its very short half-life (2 min).

Magnesium is often used for management of persistent ventricular tachycardias. It is an essential transmembrane and intracellular modulator of the electrical activity of cardiac cells. It can be administered in hypomagnesaemia, but even in normomagnesemia.

Magnesium is usually well tolerated, even in hemodynamically unstable situations; moreover, its application is rapid and simple.

Magnesium is the treatment of choice in patients with TdP tachycardia, a long QT interval and digitalis-induced cardiac arrhythmias [87]; however, its use in monomorphic VT is not so effective, with only a 30% rate of success [88].

Regarding the use of magnesium in monomorphic VT, Manz et al. addressed data from different studies and highlighted that magnesium injection should not be recommended for treatment of monomorphic ventricular tachycardia in the emergency setting [89].

As already noted, patients with structural heart disease and abnormal serum potassium concentrations at the time of an initial episode of sustained VT or VF are at high risk for recurrent VAs. Supplementation of potassium is suggested, often in combination with magnesium.

## 5. Catheter Ablation

When AADs are not sufficient to suppress VTs, CA can be considered. Vergara et al. recently published a retrospective study of 667 patients with and without ES that received catheter ablation. At 1-year follow-up, patients with ES experienced a higher risk of VT recurrence and mortality, but successful acute catheter ablation was associated with significant reduction in VT recurrence and improved 1-year survival [90].

At present, current guidelines recommend catheter ablation in patients presenting with incessant VTs or electrical storm due to VT refractory to AADs [1].

Most of the studies included CA in patients with ICM, but a similar outcome was demonstrated in NICM. Muser et al. conducted a study to compare the result of CA in electrical storm in NICM and in ICM. It was demonstrated that CA of ES was similarly effective in patients with NIDCM compared with patients with ICM, achieving reduced VA recurrence at long-term follow-up [91].

Scar-related VA substrate is currently the target of CA in different heart diseases such as NICM, arrhythmogenic right ventricular cardiomyopathy, sarcoidosis, and Chagas disease or after surgery for congenital heart disease with ventricular incision [92].

The correct timing of CA in ES is still a debated point in the literature, but there are different studies that attest that an anticipated CA could be beneficial [93,94,95].

Romero et al. studied a population of 669 patients that underwent CA ablation with a follow-up of 35 months and stratified their findings by underlying etiology of heart disease. The results demonstrated that patients referred for CA ablation with NICM and receiving multiple antiarrhythmic drug therapies had more advanced heart disease and worse overall outcomes than those patients undergoing single antiarrhythmic therapy. On the other hand, patients with ICM and multidrug failure had worse outcomes during the follow-up, primarily influenced by amiodarone failure versus failure of other AADs [93].

Frankel et al. tested the hypothesis that patients with structural heart disease and VTs were referred late for ablation and may have worse outcomes as a result. Ninety-eight consecutive patients were analyzed accordingly to the time of referral for catheter ablation; 36 patients fit the definition of early referral and 62 the late one. The procedures were successful in 89% of the patients, with a significant reduction in their VT burden. In survival analysis, the early referral group had improved VT-free survival compared to the late group (*p* = 0.01) [94].

Della Bella et al. recently demonstrated in the multicenter randomized trial “PARTITA” that CA after the first appropriate shock was associated with fewer ICD shocks, lower mortality, and a reduced risk of combined endpoints of death or worsening hospitalization for heart failure. The authors’ conclusions provided further support for considering CA of VT after the first appropriate ICD intervention as an appropriate timing [95].

## 6. Neuraxial Modulation

The autonomic system has a pivotal role in the modulation of the arrhythmic burden because the hyperactivation of the sympathetic tone is involved in the onset and maintenance of ES [96].

Conventional therapy that is normally used in ES could be a trigger of sympathetic activation: for example, a defibrillation shock can by itself further increase the sympathetic tone and the recurrence of VT or VF; obviously, epinephrine bolus used in RCP can have an adverse result such as a burst in the sympathetic tone.

Strategies to counter this trend are, therefore, necessary. In fact, neuromodulation has a crucial role in the management of electrical storm.

Sedation is the first step that can be considered. The necessity of sedation can be either a crucial point to stabilize patient hemodynamic status or for the preparation for cardioversion or a bridge to another treatment [97]. Deep sedation can be used in association with orotracheal intubation (IOT) and mechanical ventilation. Different drugs could be used, but the most used are benzodiazepine and propofol [98].

Current guidelines recommend considering deep sedation/intubation in patients with an intractable ES refractory to drug treatment. In a recent multicenter retrospective study, Martins et al. developed the first analysis of deep-sedation effectiveness in patients with ES refractory to antiarrhythmic drugs. This study considered 116 patients who received multiple electric shocks in the previous 60 (15–240) minutes before deep sedation. The drug used for sedation was a rapid-onset hypnotic agent and a rapidly acting neuromuscular blocking drug, followed by intravenous hypnotic (midazolam or propofol) and opioid agents to maintain deep sedation. Among these patients, 55 had ES termination within 15 min. Acute response to deep sedation was associated with a 55% lower risk of in-hospital death, while long-term survival did not differ between responders and nonresponders [99].

In acute settings, neuromodulation can be executed by thoracic epidural anesthesia (TEA) with infusion of a long-acting local anesthetic such as bupivacaine or ropivacaine into the T1–T2 or T2–T3 epidural space. It is a short-term method, and it is usually employed as a bridge to a more definitive solution or when catheter ablation fails. Only a few studies with small patient cohorts focusing on this topic are known; Bourke et al. demonstrated an 80% reduction in VA burden in 75% of patients unresponsive to ablation in ES [100]. In another study, 45% of patients responded to TEA [101].

Although not commonly performed, stellate ganglion blockade (LSGB) and cardiac sympathetic denervation (LCSD) are neuromodulation techniques that can be lifesaving treatment modalities and are recommended for use by the current guidelines for ES resistant to pharmacotherapy or when catheter ablation is not possible or effective.

The stellate ganglion blockade is a technique that consists of percutaneous injection of local anesthetic into the left or both stellate ganglia, blocking the efferent and afferent neurons to the heart. It is, like epidural anesthesia, a temporary measure or a bailout when catheter ablation does not work, but has good performance in the reduction of arrythmias [102]. A recent cohort study conducted on 13 patients reported that at 96 h, 62% and 92% had no VA and defibrillation episodes, respectively [103]. Using ultrasound during LSGB has been shown to greatly improve success and significantly decrease complications.

Unlike stellate ganglion blockade, cardiac sympathetic denervation is a permanent method to reduce the afferent and efferent innervation to the heart; it comprises surgical removal of the lower half of the left or bilateral stellate ganglia and the T2–T4 thoracic ganglia. In a retrospective analysis of medical records, it was pointed out that a significant reduction in the burden of implantable cardioverter-defibrillator (ICD) shocks in 90% of patients with a survival free of ICD shock was 30% in the left CSD and 48% in the bilateral CSD [104].

In addition to that, renal denervation could be promising for control of the sympathetic tone. Only limited data are available [105,106], but they seem to demonstrate that renal denervation could be an effective second-line treatment option in patients with ES and reduced LVEF when conventional catheter ablation and multiple medical treatment attempts fail to control the VTs. Its efficacy has been proved both in ICM and NICM. Several trials are still under way to test the efficacy of this method.

## 7. Bailout Therapies

In patients with hemodynamic instability despite previous interventions, more invasive options of mechanical hemodynamic support such as intra-aortic balloon pumps or extracorporeal membrane oxygenation (ECMO) can be considered [107,108].

These options could be used to create a “bridge” for different therapies such as CA or “definitive therapy” such as with a left ventricular assist device (LVAD). It has been demonstrated that early deployment of ventricular support can preserve coronary and systemic organ perfusion that can itself lead to VA suppression and prevent arrhythmic recurrences in the acute phase [107,109].

Stereotactic radioablation (SBRT) therapy has been emerging as a therapy for refractory VAs despite optimal medical therapy and CA. Ninni et al. demonstrated that SBRT significantly reduces the VT burden during ES (mean VT burden reduction: −91% [95% CI, 78–103]; *p* < 0.0001), but with a transient recurrence in the first 6 weeks, probably linked to the inflammatory response with the necessity of VT tolerance and ICD reprogramming [110].

## 8. Conclusions

ES is a medical emergency characterized by clustered VAs in a short amount of time and is associated with an increased risk of hospital admission and mortality.

Multidisciplinary management of ES in the acute phase is crucial in order to identify the arrythmia’s trigger, correct the hemodynamic instability and interrupt the VA burden.

The first-choice antiarrhythmic therapy is amiodarone, lidocaine and beta blockers (preferably not selective ones), except in patients affected by BrS and LQT syndrome when a beta agonist such as isoproterenol could be helpful. Neuromodulation and CA are actually standardized treatments for ES nonresponsive to AADs.

## Figures and Tables

**Figure 1 medicina-59-00405-f001:**
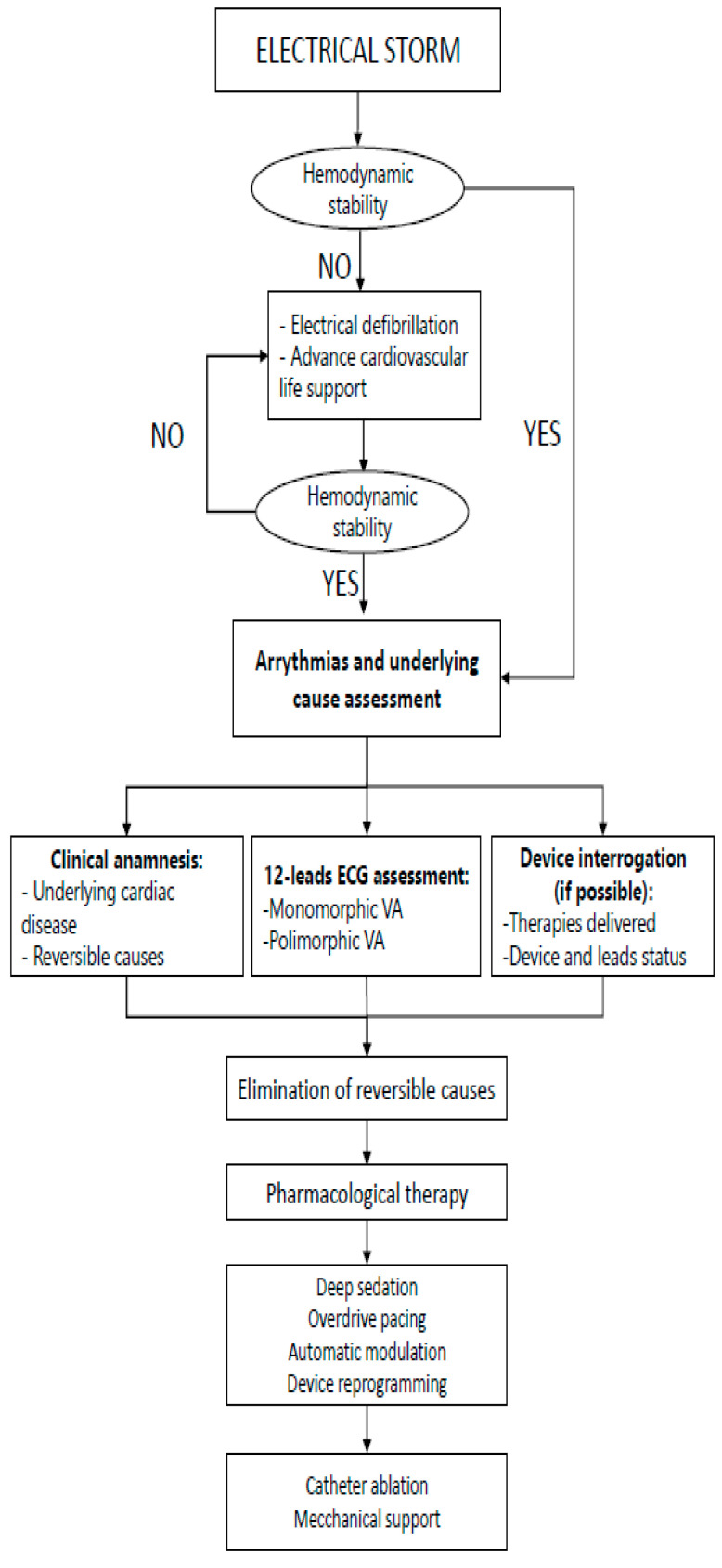
Flow chart of decision-making in ES,VA (Ventricular arrhythmias).

**Table 1 medicina-59-00405-t001:** Triggers and predictors of one-year mortality of electrical storm [32,33,34].

TRIGGERS	PREDICTORS OF ONE-YEAR MORTALITY
- Electrolyte abnormalities - Ischemia/infarction - Drug toxicity/noncompliance - Heart failure - Dysthyroidism - Alcohol and drug abuse - Fever - Sepsis - Acute starvation/dieting - Severe anemia	- Age per 10-year increase (Hazzard Ratio) HR 1.39 [1.17–1.65], *p*-value < 0.001) - Diabetes (HR 1.64 [1.07–2.51], *p*-value 0.022) - Active smoking (HR 1.65 [1.03–2.65], *p*-value 0.039) - Left ventricular dysfunction: LV end diastole diameter > 56 (HR 1.53 [0.91–2.55], *p*-value 0.11) - Right ventricular dysfunction (HR 5.61 [3.24–9.69], *p*-value < 0.001) - New York Heart Association class III (HR 1.88 [0.78–4.56], *p*-value < 0.001) or IV (HR 2.60 [0.90–7.57], *p*-value < 0.001) - Peripheral artery disease (HR 1.62 [0.93–2.83, *p*-value 0.091) - Hemoglobin at admission, per SD increase (HR 0.57 [0.47–0.69], *p*-value < 0.001) - High creatinine blood levels at admission (HR 3.87 [2.06–7.29], *p*-value ) - Use of catecholamine at admission (HR 18 [7.57–42.82], *p*-value 0.001) - Orotracheal intubation (HR 3.22 [1.73–5.99], *p*-value < 0.001)

**Table 2 medicina-59-00405-t002:** Most used medications in ES. ^a^ Timing is based on intravenous (IV) dose; if not present, it is based on oral dose, Left ventricular ejection fraction (LVEF).

Drug	Oral Dose	IV Dose	Timing ^a^	Contraindication
Quinidine	60–1600 mg Loading dose: start 200 mg every 3 h until effect (max 3 gr in first 24 h)	NO	Onset: 1–3 h Peak: 1 h Half-life: 6–8 gg	Thrombocytopenia
Mexiletine	600–1200 mg Loading dose: 400 mg initially followed by 600 mg in the first 24 h	NO	Onset: 30 min–2 h Peak: 1–4 h Half-life: 10 h	Cardiogenic shock
Lidocaine	NO	Bolus: 50–200 mg Infusion: 2–4 mg/min	Onset: 45–90 s Peak: 1–2 min Half-life: 1–2 h	Low potassium
Procainamide	NO	Bolus: 100 mg, can be repeated after 5 min if no effect (max 500–750 mg/50 mg/min) Infusion: 2–6 mg/min	Onset: 10–30 min Peak: 1 h Half-life: 2.5–5 h	Marrow failure or cytopenia
Flecainide	200–400 mg	1–2 mg/kg over 10 min	Onset: 1–5 min Peak: 2–3 h Half-life: 7–22 h	High potassium or low potassium
Propafenone	450–900 mg	Bolus: 0.5–2 mg/kg in 15–20 min), can be repeated after 60–90 min Infusion: 0.007 mg/kg/min (max 12 h)	Onset: 2 h Peak: 2–4 h Half-life: 6 h	AMI in the last 3 months and LVEF < 35%
Metoprolol	25 mg twice a day up to 200 mg/day	Bolus: 2–5 mg every 5 min up to 3 doses in 15 min	Onset: some mins Peak: 20 min Half-life:3–7 h	Severe asthma
Propanolol	80–320 mg/day	160 mg/24 h	Onset: 30 min Peak: 60–90 min Half-life: 4–5 gg	Severe asthma
Landiolol	NO	Bolus: 100 mcg/kg in 1 min Infusion: 10–40 mcg/kg/min (max 80 mcg/kg/min) max per 24 h (max 57.6 mg/kg/day)	Onset: 30 s Peak: 5 min Half-life: 4 min	Severe asthma
Esmolol	NO	Bolus: 300–500 mcg/kg in 1 min Infusion: 25–50 mg/kg/min (max 250 mg/kg/min (uptitration 5–10 min)	Onset: 60 s Peak: 5 min Half-life: 9 min	Severe asthma
Amiodarone	200–400 mg Loading dose: 600–1200 mg/24 h per 8–10 days	Loading dose: 5 mL/kg in 20 min–2 h Infusion: 600–1200 mg/24 h per 8–10 days	Onset: 10 min Peak: 15 min Half-life: 50 gg	Dysthyroidism
Sotalol	160–640 mg	0.5–1.5 mg/kg in 10 min. If necessary, it can be repeated after 6 h	Onset: 60 min Peak: 2–4 h Half-life: 10–20 h	Severe asthma
Verapamil	120–480 mg	Bolus: 5–10 mg, slow; if necessary, it can be repeated in 30 min	Onset: 1–2 min Peak: 5–15 min Half-life: 3–8 h	QRS complex tachycardia of unknown mechanism
Isoproterenol	NO	0.5–10 mcg/min	Onset: 1 min Peak:15–30 min Half-life: 2.5–5 min	Acute coronary syndromes
Magnesium sulfate	NO	Bolus: 2 gr in 5 min, can be repeated (max 4–6 g over 20 min) Infusion: 2–4 gr/h for 12–24 h	Onset: immediately Peak: in 30 min Half-life 5 h	Several renal failure, pulmonary edema

## Data Availability

Not applicable.

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
