# Peer review of "Emergency Management of Electrical Storm: A Practical Overview"

_medicina, 2023, doi:10.3390/medicina59020405_

Round 1

Reviewer 1 Report

Dear Authors,

Thank you for this work you present. You describe and review electrical storm and methods to manage this life-threatening condition in clinics. Since I am not an expert in electrical storm, I can only suggest some comments for overall improvements of the text. However, I would rather draw your attention to plagiarism matter firstly. Reference 106 published a bit over a year ago includes very similar information on the topic of the manuscript. Therefore, information in the manuscript cannot be considered significant contribution to the field. One may say it is just an unacceptable replication. Please, correct me if I am wrong.

Nevertheless, following you will find the comments to be addressed for overall improvement of the manuscript prior to considering it for publication.

1) Please, double-check and polish the text throughout. Polish the language. There is a lot of typos and minor mistakes to reconsider. Also, check use of abbreviations and specify the unspecified ones.

2) Please, organize some additional illustrations for main sections and/or subsections of the manuscript. The text seems a bit arid and boring, especially due to many abbreviations. It is hardly readable, and therefore, may and should be provided with some summarizing figures including the indicating quantities. In parallel, you may cut down on text in the correspondng parts if acceptable. For example,

2.1) you may organize a figure of risk factors for electrcal storm with odds ratio for each factor.

2.2) You may organize a figure with prognostic matter for electrical storm.

2.3) You may organize a figure for possible causes of electrical storm.

2.4) Reference to Figure 1 should not be placed in Conclusions.

2.5) Block diagram in Figure 1 seems a bit raw and poorly prepared. It is not clear what follows the exclusion of supraventricular arrhythmia and ICD shocks. Different causes of electrical storm at stable hemodynamics seems to have no influence on the therapy management. Please, rearrange to clarify decision making according to different causes (with frequencies if possible) of the storm. Also, differ the therapy methods by frequency of application. 

3) Please, check and rearrange numbering of sections and subsections if necessary. For example, there is no subsection with 3.1 number. There are two sections with number 5.

4) Please, explain the order of subsections in the Medical therapy section. I believe the subsections should be arranged with respect to class order.

5) There should be no references to literature, figures or tables in Conclusions at all. Please, move the references into appropriate sections.

6) Table 1. Please, include all medications used in electrical strom to present comprehensive information. Also, please specify IV abbreviation prior to use.

With kind regards.

Author Response

Dear Authors,

Thank you for this work you present. You describe and review electrical storm and methods to manage this life-threatening condition in clinics. Since I am not an expert in electrical storm, I can only suggest some comments for overall improvements of the text. However, I would rather draw your attention to plagiarism matter firstly. Reference 106 published a bit over a year ago includes very similar information on the topic of the manuscript. Therefore, information in the manuscript cannot be considered significant contribution to the field. One may say it is just an unacceptable replication. Please, correct me if I am wrong.

We thank the reviewer for his/her appreciation of our work and the helpful comments for improving our paper.

Even if the topic of the current manuscript could be similar to the paper cited by the reviewer (Guarracini, et al. Clinical Management of Electrical Storm: A Current Overview». Journal of Cardiovascular Medicine 22, (2021) in this manuscript we would give a really practical indications for emergency care physicians about the management of the acute phase (in particular pharmacological therapy, careful assessment of risk factors and causes) of the patient with electrical storm according recent ESC guidelines (Zeppenfeld, K., Tfelt-Hansen, J., de Riva, M. & et al. 2022 ESC Guidelines for the management of patients with ventricular arrhythmias and the prevention of sudden cardiac death Developed by the task force for the management of patients with death of the European Society of Cardiology ( ESC ) Endorsed by the. Eur. Heart J. 1–130 (2022).

Therefore in the previous manuscript published on JCM the topic was different with a broad argument on invasive strategies like catheter ablation, hemodynamic support, neuroaxial modulation and radiotherapy.  

However as suggested by the reviewer we made a further revision of the text improving different aspects drug therapy and management of triggers and causes of electrical storms.

Nevertheless, following you will find the comments to be addressed for overall improvement of the manuscript prior to considering it for publication.

Please, double-check and polish the text throughout. Polish the language. There is a lot of typos and minor mistakes to reconsider. Also, check use of abbreviations and specify the unspecified ones.

According to reviewer's suggestion, we revised the entire manuscript improving the English language and some editorial errors including abbreviations

2) Please, organize some additional illustrations for main sections and/or subsections of the manuscript. The text seems a bit arid and boring, especially due to many abbreviations. It is hardly readable, and therefore, may and should be provided with some summarizing figures including the indicating quantities. In parallel, you may cut down on text in the correspondng parts if acceptable. For example,

2.1) you may organize a figure of risk factors for electrcal storm with odds ratio for each factor.

2.2) You may organize a figure with prognostic matter for electrical storm.

2.3) You may organize a figure for possible causes of electrical storm.

2.4) Reference to Figure 1 should not be placed in Conclusions.

2.5) Block diagram in Figure 1 seems a bit raw and poorly prepared. It is not clear what follows the exclusion of supraventricular arrhythmia and ICD shocks. Different causes of electrical storm at stable hemodynamics seems to have no influence on the therapy management. Please, rearrange to clarify decision making according to different causes (with frequencies if possible) of the storm. Also, differ the therapy methods by frequency of application. 

According to reviewer's suggestion, we add a table describing about risk factors and prognostic elements of ES. Therefore we modified the diagram of figure 1.

3) Please, check and rearrange numbering of sections and subsections if necessary. For example, there is no subsection with 3.1 number. There are two sections with number 5.

According to reviewer's suggestion, we rearrange the number of the section and subsections.

4) Please, explain the order of subsections in the Medical therapy section. I believe the subsections should be arranged with respect to class order.

According to reviewer's suggestion, we rearrange the subsection with respect class order.

5) There should be no references to literature, figures or tables in Conclusions at all. Please, move the references into appropriate sections.

According to reviewer's suggestion, we erase all the references in the conclusions.

6) Table 1. Please, include all medications used in electrical strom to present comprehensive information. Also, please specify IV abbreviation prior to use.

 According to reviewer's suggestion, we improve the table with different drugs and specify IV abbreviation.

Reviewer 2 Report

1. The definition of ES remains mainly empiric since any study has examined the threshold burden of VAs or associated ICD therapies associated with increasing adverse outcome. 
Unclear, please revise.  

2. Anyways, an outcome-related definition would probably be more clinically relevant to apply therapeutic and management decisions.
"Perhaps, in stead of informal anyways would be more apropriate"

3. When polymorphic VT or 62 VF is the initiation of ES generally it is related to acute myocardial infarction and ad- 63 vanced heart failure (HF), less in ICM and NICM and only sometimes to inherited ar- 64 rhythmia syndromes, like Brugada syndrome (BrS), early repolarization syndrome (ERS), 65 short-coupled ventricular fibrillation (SCVF), long QT syndrome (LQTS), short QT syn- 66 drome (SQTS), catecholaminergic polymorphic ventricular (CPVT); in this setting it is of- 67 ten complicated by hemodynamic instability[10].
Long sentence, consider revising. 

4. "Previous history of Vas" - VAs. 

5. "Obviously, that could be explained by devices detection ability and their therapeutic role, necessary for experiencing multiple arrhythmic episodes, and also by the predisposition to 72 develop other arrhythmic episodes after the first"

Please revise. 

6. "a, ES patients have in general absolute EF reduction 76 of 3% compared to patients without ES" Please revise. 

7. ES occurring in patients with structurally normal hearts is associated with a better 98 outcome, thus ICD implantation by secondary prevention is not mandatory[16]. 
Maybe not mandatory, but in real world practice ES without clear solvable cause would mandate ICD implant even in normal EF.

8. "death reaches its edge 3" Please revise. 

9. "and 4% of patients developed electrical storm on an average of 20.6 months" Not clear, please revise.

10. "Optimal survival rates after catheter ablation". The use of "optimal" is not so optimal, please revise. 

11. "interrupt VAs with pharmacotherapy 151 in the first instance" - Please revise. 

12. "but it is not always available in all country" Please revise, quinidine is not available in lots of countries.

13. "Collateral effects" - adverse events.

14. Flecainide can be used with caution in ES even in structural heart disease when AAD/ablation does not work. Please  write few sentences about that.

15. "that magnesium could determine an increase of the cardiac index 381 during ventricular tachycardia" Please revise. 

16. "meanwhile its use in 385 monomorphic ventricular tachycardia is not so strong, with only a 30% of success" Please revise.

17. "not sufficient to deplete VTs"

18. "Vergara et all. develops a retrospective study". Please revise. 

19. "achieving a complete VAs control at long-term follow-up" This would suggest that no VA recurrences appeared, and we know this is still not possible. 

20. line 419-422 - revise, this was already stated in intro.

21. line 423-424 - bad translation, need revising the whole sentence. 

22. "support can determinate a decompression of the heart" Please revise. Also, please comment more about the LV unloading as an option for ES. Some patients are "cured" of incessant VT-a after LVAD placement. 

23. In conclusion usually we don't put references anymore. Please rewrite conclusion. 

Author Response

Response to reviewer 2 manuscript medicina-2091724

We thank the reviewer for his/her appreciation of our work and the helpful comments for improving our paper.

We modified point by point all the sentences and paragraphs of the manuscript as suggested in his/her review.

All text improvements are added in the new version of the revised manuscript submitted in the MDPI web system.

Reviewer 3 Report

In the article entitled “Emergency management of Electrical Storm: a practical overview”,

Guarracini et al. provided a review of the literature and current evidences on the management of the electrical storm. Electrical storm (ES) is a severe and life-threatening heart rhythm disorder, and patients have high risk for heart failure decompensation, in-hospital death. Antiarrhythmic drugs (AADs) are known to be the first-line management of this severe arrhythmia.

English level is sufficient.

The Authors have adequately summarized the main factors leading to this clinical scenario, and described the workflow to be adopted in order to stabilize the clinical conditions in the early phases, and manage the patient subsequently with the use of AADS and / or catheter ablation.

The references are comprehensive and up to date.

Specific comments

No major comments or issues

Minor comments:

-       Paragraph 2: Triggers causing ES could be better summarized. A table/figure would help the reader better understanding the underlying scenario potentially resulting into ES.

-       Management of ICD patients with ES could be discussed more in-depth, with a particular focus on patients having multiple shocks in case of inadequate ATP

-       Benefit of an early vs a late ablation in ES patients could be further discussed.

-       A thorough review of the manuscript to detect some typos/minor language errors should be made.

Author Response

Response to reviewer 3 manuscript medicina-2091724

In the article entitled “Emergency management of Electrical Storm: a practical overview”,

Guarracini et al. provided a review of the literature and current evidences on the management of the electrical storm. Electrical storm (ES) is a severe and life-threatening heart rhythm disorder, and patients have high risk for heart failure decompensation, in-hospital death. Antiarrhythmic drugs (AADs) are known to be the first-line management of this severe arrhythmia.

English level is sufficient.

The Authors have adequately summarized the main factors leading to this clinical scenario, and described the workflow to be adopted in order to stabilize the clinical conditions in the early phases, and manage the patient subsequently with the use of AADS and / or catheter ablation.

The references are comprehensive and up to date.

We thank the reviewer for his/her appreciation of our work and the helpful comments for improving our paper.

Specific comments

No major comments or issues

Minor comments:

-       Paragraph 2: Triggers causing ES could be better summarized. A table/figure would help the reader better understanding the underlying scenario potentially resulting into ES.

According to reviewer's suggestion, we have created a figure in the text that can better explain the role of triggers in the context of the arrhythmic storm.

-       Management of ICD patients with ES could be discussed more in-depth, with a particular focus on patients having multiple shocks in case of inadequate ATP

According to reviewer's suggestion, we improvede the section on device reprogramming in patients with arrhythmic storm, specifically about the type of reprogramming  and manuevers to be performed in this complex clinical scenario.

-       Benefit of an early vs a late ablation in ES patients could be further discussed.

According to reviewer's suggestion, we have improved the section regarding catheter ablation by focusing on the most appropriate timing.

-       A thorough review of the manuscript to detect some typos/minor language errors should be made.

      According to reviewer's suggestion, we revised the entire manuscript improving the English language and some editorial errors.

Round 2

Reviewer 1 Report

Dear Authors,

Thank you for revision of the manuscript. Unfortunately, I find your response to the previous comments insufficient to change overall recommendation. However, I have additional comments for you to address to improve the manuscript.

1) Figure 1. This is apparently a central figure of your manuscript, therefore, it must be both succinct and comprehensive, comprising all the most necessary and important information to make a clear, unambiguous decision in critical situation to minimize unnecessary risks. Although such information seems to be included in the figure, it still is poorly organized. You should organize the block diagram in a manner of clear step-by-step actions. Any additional implicit inclusion or replication of steps influencing or impeding the decision must not be allowed at the diagram.

1.1) Both versions of this figure make poor illustrative sense and have poor practical significance. Regardless of whether the hemodynamics is stable or not stable, the diagram leads to the same diagnostic blocks, and then to the same treatment blocks. You should clearly diffirentiate either diagnostics or treatment or both by hemodynamic stability. Otherwise, initial differentiation block seems to be unnecessary, although it is important. Also, you should differentiate treatment by causes identified at diagnostic stage. Further, you should differentiate unstable hemodynamics into cases of applicable and non-applicable defibrillation.

1.2) Please, specify IOT abbreviation prior to use.

2) Tables. 

2.1) It is recommended to include some statistical quantitaties to try and differentiate predictors by practical significance in Table 1.

2.2) It is recommended to include all relevant predictors in Table 1.

2.3) Please, correct the numbering of tables.

3) References. Please, pay particular attention and correct numbering of some references in the text. For instance, there is no reference with 112 number, yet it appears several times in the text. On the contrary, some references (e.g., 107 and 108) are not addressed in the text at all. 

4) Check and polish remaining typos and minor mistakes.

With kind regards.

Author Response

Dear Authors,

Thank you for revision of the manuscript. Unfortunately, I find your response to the previous comments insufficient to change overall recommendation. However, I have additional comments for you to address to improve the manuscript.

We thank the reviewer for his/her for helpful comments for improving further our paper.

1) Figure 1. This is apparently a central figure of your manuscript, therefore, it must be both succinct and comprehensive, comprising all the most necessary and important information to make a clear, unambiguous decision in critical situation to minimize unnecessary risks. Although such information seems to be included in the figure, it still is poorly organized. You should organize the block diagram in a manner of clear step-by-step actions. Any additional implicit inclusion or replication of steps influencing or impeding the decision must not be allowed at the diagram.

1.1) Both versions of this figure make poor illustrative sense and have poor practical significance. Regardless of whether the hemodynamics is stable or not stable, the diagram leads to the same diagnostic blocks, and then to the same treatment blocks. You should clearly diffirentiate either diagnostics or treatment or both by hemodynamic stability. Otherwise, initial differentiation block seems to be unnecessary, although it is important. Also, you should differentiate treatment by causes identified at diagnostic stage. Further, you should differentiate unstable hemodynamics into cases of applicable and non-applicable defibrillation.

1.2) Please, specify IOT abbreviation prior to use.

According to reviewer's indications, we modified the figure 1 of the manuscript.

2) Tables. 

2.1) It is recommended to include some statistical quantitaties to try and differentiate predictors by practical significance in Table 1.

2.2) It is recommended to include all relevant predictors in Table 1.

2.3) Please, correct the numbering of tables.

According to reviewer's indications, we modified the table 1 of the manuscript.

3) References. Please, pay particular attention and correct numbering of some references in the text. For instance, there is no reference with 112 number, eti t appears several times in the text. On the contrary, some references (e.g., 107 and 108) are not addressed in the text at all.

According to reviewer's indications, we corrected the references in the manuscript and in the bibliography.

 4) Check and polish remaining typos and minor mistakes.

According to reviewer's suggestion, we revised the entire manuscript improving the English language and some editorial errors including abbreviations.